# Perioperative Risk Factors for Bleeding in Adolescents Undergoing Pedicle Screw Instrumentation for Scoliosis

**DOI:** 10.3390/children10020381

**Published:** 2023-02-15

**Authors:** Venla Soini, Johanna Syvänen, Ilkka Helenius, Linda Helenius, Arimatias Raitio

**Affiliations:** 1Department of Paediatric Surgery and Paediatric Orthopaedic Surgery, University of Turku and Turku University Hospital, Savitehtaankatu 5, 20520 Turku, Finland; 2Department of Surgery, Vaasa Central Hospital, Wellbeing Services County of Ostrobothnia, 65130 Vaasa, Finland; 3Department of Orthopaedics and Traumatology, University of Helsinki and Helsinki University Hospital, 00029 Helsinki, Finland; 4Department of Paediatric Orthopaedic Surgery, Helsinki New Children’s Hospital, 00290 Helsinki, Finland; 5Department of Anaesthesiology and Intensive Care, Turku University Hospital and University of Turku, 20521 Turku, Finland

**Keywords:** bleeding, blood loss, hidden blood loss, pediatric orthopedics, scoliosis, neuromuscular scoliosis, idiopathic scoliosis, spinal fusion

## Abstract

Progressive scoliosis eventually leads to extensive spinal fusion surgery, which carries a risk for significant bleeding. Neuromuscular scoliosis (NMS) patients have an additional inherent risk of major perioperative bleeding. The purpose of our research was to investigate the risk factors for measured (intraoperative, drain output) and hidden blood loss related to pedicle screw instrumentation in adolescents, divided into adolescent idiopathic scoliosis (AIS) and NMS patient groups. A retrospective cohort study with prospectively collected data of consecutive AIS and NMS patients undergoing segmental pedicle screw instrumentation at a tertiary level hospital between 2009 and 2021 was conducted. In total, 199 AIS (mean age 15.8 years, 143 females) and 81 NMS patients (mean age 15.2 years, 37 females) were included in the analysis. In both groups, levels fused, increased operative time, and smaller or larger size of erythrocytes were associated with perioperative blood loss (*p* < 0.05 for all correlations). In AIS, male sex (*p* < 0.001) and the number of osteotomies correlated with more drain output. In NMS, levels fused correlated with drain output, *p* = 0.00180. In AIS, patients’ lower preoperative MCV levels (*p* = 0.0391) and longer operation times, *p* = 0.0038, resulted into more hidden blood loss, but we did not find any significant risk factors for hidden blood loss in NMS patients.

## 1. Introduction

Scoliosis is a common problem in pediatric population. If severe scoliosis is left untreated, it leads to difficulty in maintaining posture, skin problems, restrictively reduced lung function, and increased mortality [1]. In adolescent idiopathic scoliosis (AIS), a conservative treatment with bracing is proven to be effective, but when the progression of a curve occurs despite adequate treatment, spinal fusion is chosen to prevent further progression [2]. In neuromuscular scoliosis (NMS), progression cannot be controlled with conservative methods, and that often leads to extensive thoracolumbar spinal fusion [3,4]. Spinal fusion is a major orthopedic operation and includes a high risk for voluminous intraoperative blood loss [5,6,7]. Neuromuscular patients typically require more extensive fusion than patients with AIS [8,9,10] and have been proven to have more intra- and postoperative bleeding per levels fused as compared to AIS patients [10,11,12,13]. Drain output often equals the measured intraoperative blood loss in patients with AIS [11,12,14,15]. In a recent study, hidden blood loss in addition to measured blood loss (intraoperative and drain output) contributed to 30% of the total perioperative bleeding in NMS [16].

The risk factors for intraoperative bleeding for AIS and NMS patients are reasonably well understood, while for drain output and hidden blood loss, less is known for both groups. Given the relatively high amount of both drain output and hidden blood loss in scoliosis surgery, it would be important to understand their risk factors. In previous studies, longer operative time, lower BMI, more extensive fusion, larger preoperative cobb angle for main curve, and platelet levels have played roles in more extensive bleeding concerning AIS patients [17,18,19,20,21]. In comparison, in NMS patients, lower BMI, higher age, larger preoperative main curvature, preoperative hematocrit, older age, and number of osteotomies have been described to influence more voluminous intraoperative bleeding [22,23,24,25]. 

The aim of our study was to determine the perioperative risk factors for blood loss in children with scoliosis undergoing posterior spinal fusion with segmental pedicle screw instrumentation. We hypothesized that preoperative low hematocrit and platelet values would be independent risk factors for perioperative bleeding.

## 2. Materials and Methods

We present a retrospective study on the risk factors for perioperative blood loss for children and adolescents undergoing instrumented posterior spinal fusion using pedicle screws. Due to the diversity of scoliosis etiologies and differences in the extents of the operations, our patients were divided into groups of AIS and NMS. Patients with other types of scoliosis, such as syndromic or congenital scoliosis, were excluded. The operations were conducted at a single tertiary hospital center between the years 2009 and 2021. The data collection was carried out prospectively through our institutional pediatric spine register. Perioperative surgical and anesthetic protocols were standardized including pre- intra- and postoperative care. Preoperative examinations consisted of clinical examination, imaging with anteroposterior and lateral radiographs, and MRI imaging for patients with AIS, consultations (anesthesiology and pediatric neurology for NMS), and surgical planning (implant placement, levels fused, and osteotomies). The intraoperative anesthetic protocol with preset rates of infusions (tranexamic acid 30 mg/kg iv bolus and infusion 10 mg/kg/h during the procedure), and thresholds for vitals and transfusions were maintained. The surgical protocol included patient positioning; exposure with electrocautery; hemostasis; pedicle screw insertion with imaging verification; curve correction with titanium rods, and direct translation and segmental compression/distraction (CD Legacy and Solera 5.5/6.0, Medtronic International, USA); local bone autografting from facetectomies; bone graft extender (AIS, Grafton, Medtronic, USA); allogeneic bone grafting (NMS) and gelatin matrix with human thrombin (Floseal, Baxter, USA); and a multilayer wound closure with single subfascial drain insertion. Postoperatively, all patients were admitted to an intensive care unit for the first 24 postoperative hours, after which the drainage was removed. All operations were performed by the same experienced orthopedic spine surgeon (I.H.). The methods of our standardized perioperative protocol are described in more detail in our previous papers [15,16,26]. Approval from ethical committee was granted for the study (ETMK 96/1801/2020). 

We analyzed the risk factors for visible bleeding by measuring intraoperative blood loss (IBL) and 24 h drain output (DBL). For hidden blood loss (HBL), an estimate was calculated using the principles demonstrated in the previous literature [27,28,29]. In addition, we analyzed risk factors for total blood loss (TBL), combining intraoperative, drain output, and hidden blood loss. 

The factors included in the bivariate analysis were sex, age, BMI, preoperative main curvature, and preoperative laboratory testing levels (leukocytes, erythrocytes, hemoglobin, hematocrit (HCT), mean corpuscular volume (MCV), mean corpuscular hemoglobin (MCH), platelets, C-reactive protein (CRP), ferritin, international normalized ratio (INR), activated partial thromboplastin clotting time (aPTT), thrombin time (TT), procalcitonin (PCT), creatinine kinase (CK), creatinine, potassium (K), sodium (Na) and magnesium (Mg)).

A bivariate analysis was performed for each type of bleeding (mL) and each possible risk factor individually in both groups. Normal distribution was statistically verified, and non-parametric data were analyzed with the Wilcoxon rank sum test. In the bivariate analysis performed for each type of bleeding and AIS/NMS group individually, multiple potential risk factors were identified. These are described in detail below, divided into three classes: preoperative, perioperative, and postoperative factors. For the factors, visual and statistical correlation was observed in a bivariate analysis, and a multiple linear regression analysis was performed. The results are represented with correlation coefficients and *p*-values for each factor and bleeding type, and a threshold of <0.05 was maintained for significance. The factors resulting in non-significant results for all types of bleeding were excluded from the results for clarity. All analyses were carried out using JMP Pro 16.2.0. 

Our main outcome measures were risk factors for drain output and hidden blood loss (mL) in adolescent patients undergoing spinal fusion for scoliosis. The secondary outcome included risk factors for intraoperative and total blood loss. 

## 3. Results

In total, 280 consecutive patients were included in the analysis (199 AIS and 81 NMS). The basic demographics are reported in Table 1. The preoperative laboratory results did not significantly differ between the groups except for a difference in creatinine levels, which were significantly higher in the AIS patients (*p* < 0.001, Table 2). 

### 3.1. Preoperative Factors

In the AIS group, male sex, larger preoperative main curve, higher erythrocytes, ferritin, aPTT, creatinine, and potassium and lower platelets have statistically significant positive correlations in the bivariate analysis on intraoperative bleeding. Similarly, male sex; larger main curve; and the laboratory results of erythrocytes, hemoglobin, hematocrit, and creatinine were potential risk factors for drainage bleeding. Larger BMI, higher erythrocyte, and lower MCV showed potential influences on hidden blood loss. Male sex; larger BMI; preoperative main curves; laboratory levels of higher erythrocytes, hemoglobin, hematocrit, ferritin, and creatinine; and lower MCV levels were correlated with total bleeding. 

In a bivariate analysis of the NMS group, older age and higher preoperative ferritin levels were potential risk factors for intraoperative bleeding, and older age and higher Mg levels correlated positively with drainage bleeding. Male sex was a potential risk factor for hidden blood loss. Male sex; older age; and preoperative hemoglobin, hematocrit, MCV, MCH, ferritin, and potassium levels were all identified as risk factors for total bleeding. 

### 3.2. Perioperative Factors

Concerning perioperative factors, longer operative time was a potential risk factor for all types of bleeding in AIS and for intraoperative and total bleeding in the NMS group. The number of levels fused had a correlation to intraoperative, drainage, and total bleeding in patients with AIS and drain output in the NMS group. Osteotomies were a potential risk factor for drainage bleeding in AIS patients. 

### 3.3. Multiple Linear Regression Analysis

Concerning AIS patients’ intraoperative bleeding, male sex (0.399, *p* = 0.0044), number of levels fused (0.837, *p* < 0.001), and operative time (0.243, *p* < 0.001) were all individual risk factors in the multivariable analysis. Male sex (0.2440, *p* = 0.0227) and number of osteotomies (0.912, *p* = 0.0231) showed significant correlations to drainage bleeding. MCV levels (−0.094, *p* = 0.0391) and operative time (0.221, *p* = 0.0038) correlated with hidden blood loss. Male sex (0.353, *p* < 0.001), BMI (0.525, *p* < 0.001), fused levels (0.688, *p* < 0.001), and operative time (0.181, *p* < 0.001) were risk factors for total bleeding (Table 3). 

In the NMS group, preoperative ferritin (0.071, *p* = 0.0165) and longer operative time (0.931, *p* < 0.001) were associated with increased intraoperative bleeding. The number of levels fused correlated with drainage bleeding (0.997, *p* = 0.0180), while higher preoperative MCV levels (0.981 *p* < 0.001) and operative time (0.223, *p* < 0.001) correlated with more total bleeding (Table 4). There were no significant risk factors for hidden blood loss in the NMS group. 

### 3.4. Postoperative Factors

Concerning postoperative factors, a few patients were treated without postoperative drainage, for example, because of cerebrospinal fluid leak or other complication (N = 20 in AIS and N = 13 in NMS). These patients had significantly larger hidden blood loss compared to patients with postoperative drainage in both groups, *p* < 0.001 for AIS and *p* = 0.0096 for NMS. However, there were no differences between patients treated with or without drainage concerning total bleeding in both groups (*p* = 0.3044 for AIS and *p* = 0.6184 for NMS).

## 4. Discussion

Based on the results of our analysis, the preoperative laboratory levels of AIS and NMS patients differed significantly only when comparing creatinine levels, as expected due to the nature of neuromuscular diseases having less muscle mass. In both groups, levels fused, increased operative time, and smaller or larger size of erythrocytes were associated with more perioperative bleeding. In contrast to our hypothesis, preoperative laboratory findings such as low levels of platelets or hematocrit were not significant risk factors for intraoperative, drain output, or hidden blood loss in children with AIS or NMS. 

Our aim was to identify risk factors for hidden blood loss and drainage blood loss in both groups. In the AIS group, male sex and larger number of osteotomies correlated with more drainage bleeding, and increasing operative time and lower MCV levels resulted in less hidden blood loss. Concerning NMS patients, fusion level was an independent risk factor for more voluminous drainage bleeding, but no independent risk factors for hidden blood loss could be found based on our data. 

Regarding intraoperative bleeding, operative time was a risk factor for both NMS and AIS. In AIS patients only, male sex and fusion level had impacts on intraoperative bleeding, and in NMS, a higher preoperative ferritin level was an independent risk factor for more voluminous bleeding. Male sex, larger BMI, longer operation, and more levels in fusion led to more total bleeding in AIS, as higher MCV levels and longer posterior time correlated with increasing total bleeding in NMS. 

### 4.1. Comparison to Literature

Brenn et al. evaluated the differences in clotting parameters between NMS and AIS patients and found a significant difference in partial thromboplastin time [30]. Fernandez et al. stated that preoperative coagulative function tests predict more voluminous intraoperative bleeding [31]. In our preoperative laboratory levels, such differences in preoperative bleeding parameters could not be observed. 

In their study of 1859 AIS patients, Song et al. divided their dataset into massive (>30% of estimated blood volume) and non-massive (<30% EBV) blood loss groups and found that longer operation duration, more number of fusion levels, lower BMI, larger preoperative Cobb angle, lower preoperative platelet count, and increasing bleeding time (INR) were independent risk factors for massive blood loss [19]. In a similar setting with NMS patients, Maio et al. found that lower BMI was a risk factor for massive blood loss associated with spinal fusion [24]. Additionally, Jia et al. conducted a similar study of massive blood loss in NMS and discovered that a greater number of fusion levels, a BMI lower than 16.8, an age greater than 15 years, and a duration longer than 4.4 h were all risk factors for massive blood loss [22]. Our primary multivariable analysis supports the effect of operative time to the intraoperative bleeding in both groups and the influence of a greater fusion level in more voluminous bleeding in AIS. In addition, the more excessive curvature in AIS and older age in NMS were also seen as possible risk factors in our bivariate analysis. On the contrary to Song et al., in our AIS dataset, BMI correlated positively with total bleeding. To enable more accurate comparison to these previous studies, we also analyzed our data by splitting the AIS and NMS groups further into massive and-non massive groups, and in our dataset, 11 (5.5%) patients in the AIS group and 36 (44%) patients in the NMS group exceeded 30% of EBV. When comparing these two groups to our demographic factors, in the AIS group, a greater number of levels fused (*p* = 0.0036), younger age (*p* = 0.0067), and higher INR (*p* = 0.0157) were possible independent risk factors for massive blood loss. In the NMS group, there were no independent risk factors for massive blood loss. However, in our opinion, comparing the continuous data instead of grouping provides a better overall estimate of the risk factors.

Li et al. used a similar grouping setting as mentioned above and compared drainage bleeding to categorized risk factors with threshold limits [32]. They found a correlation between low BMI, large curvature angle, number of fused levels, and use of osteotomies as risk factors for drainage bleeding in AIS. Choi et al. assessed the influencing factors for more voluminous drainage bleeding in AIS patients and found that number of osteotomies correlated with more drainage bleeding in their 50-patient cohort [33]. Our cohort consisted of 199 AIS patients and is in line with these previous observations concerning the effect of osteotomies on drain output. Guay et al. found no significant risk factors for drainage bleeding, yet there was a correlation between postoperative bleeding and the length of time the drainage was in place [34]. In our surgical protocol, the drain was removed routinely at 24 h postoperatively to avoid risk of possible infection [35], so this difference could not be observed. 

Kolz et al. assessed the hidden blood loss and its risk factors in their dataset of 67 AIS patients undergoing spinal fusion and found that BMI > 25 was a risk factor in their categorized multivariable analysis for HBL in spinal fusion [36]. Longer operative time (>3.5 h) and higher age (>14) were also identified as risk factors in the bivariate analysis. Our dataset of AIS patients is three times as big, used continuous data instead of categorical variables, and showed somewhat different results, as a correlation between HBL and MCV and longer operative time were observed in a multivariable analysis. 

Li et al. studied the risk factors for hidden blood loss in AIS in their review and found that the number of fused levels was a risk factor for hidden blood loss [18]. Wang et al. presented HCT as an independent risk factor for hidden blood loss in AIS [27]. The current study did not reveal correlations between hidden blood loss and preoperative HCT or number of levels fused in AIS patients. Bai et al. studied the predictive factors for hidden blood loss in spinal surgery for patients with scoliosis. They found that receiving autologous and allogenic transfusions were risk factors for hidden blood loss in multivariable analysis [37]. This was not originally included but we ran an additional analysis to determine this factor, and in our dataset, receiving transfusions did not correlate to hidden blood loss, neither in the multivariable nor in the bivariate analyses.

Tang et al. found a correlation between preoperative hemoglobin level and intraoperative blood loss in AIS [21]. Lewen et al. observed the relationship between preoperative hematology laboratory results and intraoperative blood loss in spinal fusion in NMS patients. They found that higher preoperative hematocrit and lower platelet count increased intraoperative blood loss [23]. In current study, lower platelets were potential risk factors for greater intraoperative bleeding in the AIS group, but none of these laboratory results correlated with the amount of intraoperative bleeding in the multivariable regression analysis. Additionally, Hb and HCT showed a correlation in the bivariate analysis to total blood loss in both the AIS and NMS groups. 

Ialenti et al. found that male sex and operative time were risk factors for more voluminous bleeding in overall perioperative bleeding in AIS patients, which is in line with our findings, as male sex was a significant risk factor for intraoperative, drainage, and total bleeding [20].

Toombs et al. conducted a study on 57 NMS patients undergoing spinal fusion and found that older age and longer operative time were risk factors for increased intraoperative bleeding [25]. Our results support these findings regarding the operative time and intraoperative bleeding, *p* < 0.001, but in our dataset, age had a significant correlation with bleeding only in the bivariate analysis. In their study, Toombs et al. described age means of 16.6 years, whereas in the current cohort, age mean was slightly lower, 15.2 years. Furthermore, in our analysis, a higher ferritin level correlated with more bleeding together with posterior time in the multivariable regression analysis, but in Toombs et al.’s analysis, ferritin was not included. This might explain the effect with age, as an upward trend in ferritin levels in adolescent males over 14 years old has been observed in previous studies [38]. Additionally, it should be noted that their dataset also consisted of patients with a combined anterior–posterior operative technique. Therefore, the results are not entirely comparable to our data with a standardized posterior spinal fusion protocol. 

The effect of antifibrinolytics on bleeding in spinal fusion for scoliosis has been studied extensively in recent years. Ahlers et al. and Verma et al. found that antifibrinolytics such as tranexamic acid and epsilon-aminocaproic (EACA) reduce the amount of bleeding, especially drainage bleeding, but did not have an effect on transfusion rate [39,40]. Shapiro et al. studied the benefits of TXA for bleeding in spinal fusion in NMS children [41]. On the contrary, Bosch et al. found, in their prospective cohort study, that tranexamic acid reduced the fibrinolysis and need for transfusions in posterior spinal fusion for AIS but did not reduce EBL [42]. Thompson et al. found a decrease in bleeding in neuromuscular scoliosis patients treated with aminocaproic acid [43]. Dhawale et al. stated that tranexamic acid was more efficient than EACA [44]. The effects of antifibrinolytics was not included in our analysis as we have decided to give tranexamic acid to all of our patients as a part of our standardized protocol (30 mg/kg, max 1500 mg as a bolus and intraoperative infusion of 10 mg/kg/h max 500 mg/h).

### 4.2. Strengths and Limitations

Concerning the analysis of risk factors for perioperative bleeding, our standardized surgical and, more importantly, anesthetic protocols increase the reliability of the current findings. Furthermore, prospective data collection improves the chances of creating high-quality patient data. The number of AIS and NMS patients was adequate for multivariable analyses.

This is the first article investigating risk factors for hidden blood loss and drainage bleeding related to spinal fusion surgery in NMS patients. It also provides more information about the risk factors in AIS patients undergoing spinal fusion.

The retrospective analysis of prospectively collected data is its main limitation. Additionally, the usage of intraoperative products affecting bleeding such as tranexamic acid could not be included in the current analysis, since all our patients received 30 mg/kg as a bolus and an infusion of 10 mg/kg/hour as a part of our standardized protocol.

## 5. Conclusions

In our retrospective cohort study of 280 adolescents with a standardized perioperative protocol, male sex and a higher number of osteotomies increased the risk of more voluminous drainage bleeding in AIS patients in the multivariable regression analysis. Longer operative time and lower preoperative MCV had an increasing impact on hidden blood loss in the AIS group. Number of levels fused was an independent risk factor for drainage bleeding in the NMS patients. No independent risk factors for hidden blood loss in NMS patients could be found based on our data.

## Figures and Tables

**Table 1 children-10-00381-t001:** Patient demographics [16].

Demographics	AIS (N = 199)	NMS (N = 81)	*p* Value
Age	15.6 ± 2.1	15.2 ± 3.4	*p* = 0.201
Sex (M/F)	56/143	44/37	*p* < 0.001
Major curve, degrees Preoperatively Postoperatively	52 ± 8 12 ± 5	72 ± 18 20 ± 12	*p* < 0.001 *p* < 0.001
Pelvic instrumentation, n (%)	0/0	63 (77) %	
Operative time (h)	3.1 ± 0.07	4.2 ± 0.13	*p* < 0.001
Intraoperative bleeding (mL)	554 ± 349	1085 ± 1049	*p* < 0.001
Drain output (mL)	489 ± 188	566 ± 208	*p* = 0.0147
Hidden blood loss (mL)	398 ± 411	566 ± 533	*p* = 0.0332
Total bleeding (mL)	1358 ± 544	1914 ± 1006	*p* < 0.001

**Table 2 children-10-00381-t002:** Preoperative laboratory levels.

Laboratory Levels	AIS	NMS	*p*-Value Difference
Erythrocytes (x10E12/L)	4.78 ± 0.41	4.8 ± 0.4	0.9864
Hemoglobin (g/L)	138 ± 11	138 ± 14	0.9060
Hematocrit (HCT)	0.41 ± 0.03	0.41 ± 0.04	0.6689
MVC (fl)	86 ± 3	86 ± 6	0.6781
MCH (pg)	29 ± 1	29 ± 3	0.6778
Platelets (x10E9/L)	268 ± 57	273 ± 87	0.8273
Ferritin (μg/L)	38 ± 26	48 ± 43	0.1546
aPTT (s)	29.0 ± 3.9	28 ± 6.6	0.8531
Creatinine (μmoL)	66 ± 23	38 ± 18	**<0.0001**
Potassium (K) mmoL/L	4.1 ± 0.3	3.9 ± 0.3	0.0843
Magnesium (Mg) (mmoL/L)	0.89 ± 0.12	0.82 ± 0.08	0.3486

**Table 3 children-10-00381-t003:** Multiple linear regression analysis for each type of bleeding in AIS, and correlation coefficients and *p*-values for each significant factor and bleeding type.

	IBL	DBL	HBL	TBL
*Preoperative factors*				
Sex	0.3992, *p* = 0.0044	0.2440, 0.0227		0.3530, <0.001
BMI				0.5254, <0.001
MCV			0.9909, 0.0391	
*Perioperative factors*				
Fused vertebrae	0.8373, <0.001			0.1806, <0.001
Operative time	0.2425, <0.001		0.2211, 0.0038	0.6884, <0.001
Osteotomies		0.9118, 0.0231		

**Table 4 children-10-00381-t004:** Multiple linear regression analysis for each type of bleeding in NMS, and correlation coefficients and *p*-values for each factor and bleeding.

	IBL	DBL	HBL	TBL
*Preoperative factors*				
MCV				0.9807, <0.001
Ferritin	0.0714, *p* = 0.0165			
*Perioperative factors:*				
Fusion level		0.9971, 0.0180		
Operative time	0.9305, <0.001			0.2227, <0.001

## Data Availability

Please contact author for data requests.

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
