# Peer review of "Perioperative Risk Factors for Bleeding in Adolescents Undergoing Pedicle Screw Instrumentation for Scoliosis"

_children, 2023, doi:10.3390/children10020381_

Round 1
Reviewer 1 Report
This study looks at the preoperative risk factors for blood loss in scoliosis surgery.
The topic is original and relevant to spinal surgeons. It add to the literature.
Limitation- as stated by author - retrospective study.
Conclusion is consistent with evidence and written paper.
The references are appropriate.
This paper is well written with good English.
The introduction and methods is well described.
The results is well presented.
The discussion is good with description of other studies.
The conclusion is consistent with the aim of the paper
Author Response
We thank Reviewer 1 for the positive comments regarding our manuscript.
Reviewer 2 Report
* The subject is somewhat clear, and it has been explored much more than the current introduction gives credit. The article presents a good idea. Although the initial question is interesting.
* The authors discussed the limitation of this study.
* The conclusion is not justified by the methods and the results. Please reformulate.
